# Extreme slow growth as alternative strategy to survive deep starvation in bacteria

Declan A. Gray[1], Gaurav Dugar [ID] [2], Pamela Gamba[1], Henrik Strahl[1], Martijs J. Jonker[3] & Leendert W. Hamoen [ID] [2]

Bacteria can become dormant or form spores when they are starved for nutrients. Here, we find that non-sporulating *Bacillus subtilis* cells can survive deep starvation conditions for many months. During this period, cells adopt an almost coccoid shape and become tolerant to antibiotics. Unexpectedly, these cells appear to be metabolically active and show a transcriptome profile very different from that of stationary phase cells. We show that these starved cells are not dormant but are growing and dividing, albeit with a doubling time close to 4 days. Very low nutrient levels, comparable to 10,000-fold diluted lysogeny broth (LB), are sufficient to sustain this growth. This extreme slow growth, which we propose to call 'oligotrophic growth state', provides an alternative strategy for *B. subtilis* to endure nutrient depletion and environmental stresses. Further work is warranted to test whether this state can be found in other bacterial species to survive deep starvation conditions.

[1] Centre for Bacterial Cell Biology, Institute for Cell and Molecular Biosciences, Newcastle University, Baddiley-Clark Building, Newcastle upon Tyne NE2 4AX, UK. [2] Swammerdam Institute for Life Sciences, University of Amsterdam, Science Park 904, 1098 XH Amsterdam, The Netherlands. [3] MicroArray Department and Integrative Bioinformatics Unit, Swammerdam Institute for Life Sciences, University of Amsterdam, Science Park 904, 1098 XH Amsterdam, The Netherlands. Correspondence and requests for materials should be addressed to L.W.H. (email: l.w.hamoen@uva.nl)

**B**acteria encounter multiple environmental stresses during their life, including depletion of nutrients. Some genera, such as *Bacillus*, *Streptomycetes*, and *Clostridia*, have developed specialized cells, spores, to survive extended periods of nutrient depletion. These dormant spores contain a dehydrated cytoplasm encased in a highly protective multilayer spore coat, making them resistant to extreme environmental conditions[1]. However, the majority of bacterial species do not form spores but are nevertheless able to survive prolonged periods of nutrient starvation. For example, the fish pathogen *Flavobacterium columnare* remains viable after 14 days of incubation in pure water[2]. *Brucella suis*, the causative agent of swine brucellosis, can survive 6 weeks of incubation in a salt solution[3], and *Escherichia coli* can withstand 260 days of incubation in river water[4]. It should be mentioned that in all these cases it was only a small fraction of the population that survived. Cells that are exposed to deep starvation conditions typically show morphological changes, e.g. coiling in the case of *F. columnare* cells[2], and cell shrinkage in the case of *Micrococcus luteus* and *Staphylococcus aureus*[5,6].

A recurring question is how dormant these long-term starved cells are. In contrast to bacterial spores, which are fully dormant, it is likely that starved non-sporulating cells have to maintain some basic cellular activities, such as the proton motive force, to remain viable. For example, *S. aureus* cells starved for 7 days showed some sensitivity toward chloramphenicol indicating ongoing translation[6]. On the other hand, *Mycobacterium tuberculosis* starved for 6 weeks tolerated extensive treatment with the RNA polymerase inhibitor rifampicin or the mycobacterial cell wall synthesis inhibitor isoniazid, suggesting a fully dormant state[7].

The soil bacterium *Bacillus subtilis* forms dormant endospores upon prolonged nutrient starvation. Sporulation is a costly differentiation process in terms of time and energy, and cannot be reversed once the asymmetric sporulation septum has been formed[8,9]. That is why *B. subtilis* only initiates sporulation in a fraction of cells in a population[10]. This differentiation bifurcation is known as a bet-hedging strategy, as it enables the population to survive when starvation continues or to quickly respond when there is an influx of fresh nutrients[11,12]. However, this bifurcation raises the question what happens with the non-sporulating cells when the starvation period continues.

In this study we show that non-sporulating *B. subtilis* cells can survive for many months in pure water, and that they become tolerant to different stresses. Using cell biological techniques and a novel assay for growth, we were able to demonstrate that these cells are not dormant but instead are growing slowly. Transcriptome profiles of these cells differed substantially from exponentially growing and stationary phase cells, indicating that these cells undergo an alternative cellular adaptation process. We propose to call this the "oligotrophic growth state". The advantage of this cellular differentiation over sporulation and whether oligotrophic growth is a common mechanism in bacteria to survive prolonged nutrient depletion are further discussed.

## Results

**Survival of non-sporulating *B. subtilis*.** To facilitate the examination of non-sporulating *B. subtilis* cells, we made use of a sporulation-deficient mutant. Sporulation begins with phosphorylation of the response regulator Spo0A[13]. Since this transcription factor regulates many other stationary phase processes, including biofilm formation, genetic competence, and degradative enzyme production[14], we left the *spo0A* gene intact and instead used a strain with an impaired *spoIIE* gene, which is one of the first essential sporulation genes induced by Spo0A, and is not required for other differentiation processes[15]. The Δ*spoIIE* strain

was grown in Spizizen minimal medium (SMM) at 37 °C under continuous shaking. Samples were withdrawn at regular time intervals to determine viability by measuring colony-forming units (CFU). Unexpectedly, this non-sporulating strain not only survived several days without fresh nutrients, but even after 100 days the culture still contained some viable cells that formed colonies (Fig. 1a).

In the stationary growth phase unused amino acids are left in the medium and substantial levels of overflow metabolites accumulate. Moreover, we observed a drop in optical density and in CFUs during the first 2 days of starvation (Fig. 1a),

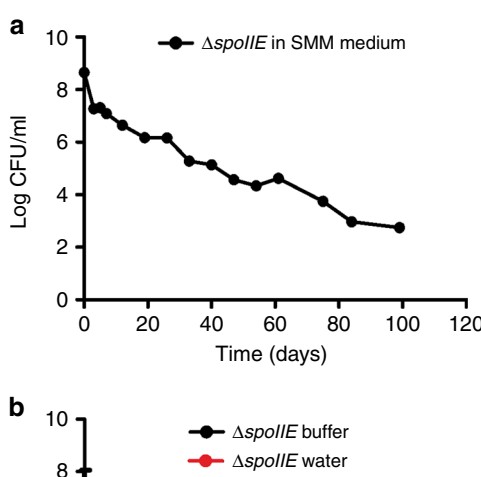

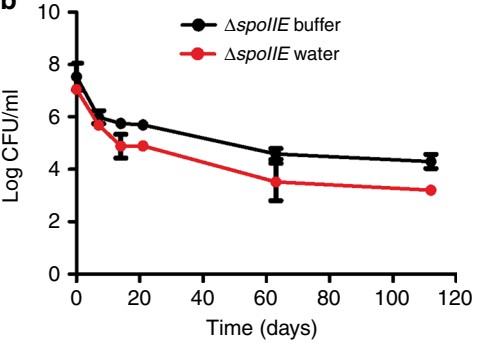

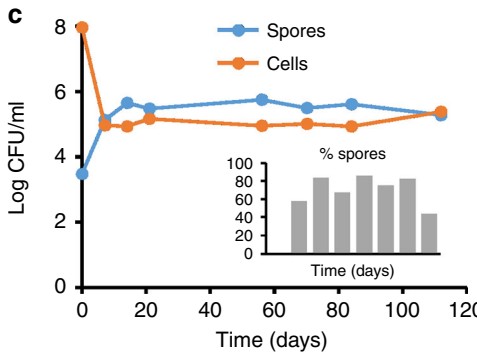

**Fig. 1** Long-term survival of non-sporulating *B. subtilis*. **a** Colony-forming units (CFU) of *B. subtilis* Δ*spoIIE* (strain DG001) incubated in Spizizen minimal medium (SMM). **b** CFU of Δ*spoIIE* cells that were first grown for 2 days in SMM, and subsequently filtered and incubated in either starvation buffer or water (=0 days time point). The CFU numbers of the first time point are therefore comparable to those of time point 2 days in graph (**a**). Averages and standard deviation from three independent experiments are depicted. The difference between the two graphs becomes significant after day 7 ($p < 0.05$, unpaired two-tailed *t*-test). **c** CFU of spores and cells in a wild-type *B. subtilis* culture (strain BSB1) incubated in starvation buffer. The percentage of spores is indicated in the bar diagram. Results of two replicate experiments are shown in Supplementary Figure 2A. See Methods for details on growth and starvation conditions

suggesting a release of more nutrients. To be sure that these nutrients could not be used for further growth, cultures were filtered after 2 days of stationary growth and subsequently resuspended in a modified SMM lacking any carbon source, here called starvation buffer. This deep starvation procedure, schematically outlined in Supplementary Figure 1, was used in all further experiments. In this starvation buffer, deprived of any nutrients, non-sporulating *B. subtilis* cells still survived for more than 100 days (Fig. 1b). The pH remained stable for at least the first 14 days, which is not surprising considering the presence of 100 mM phosphate buffer. However, even when cells were resuspended in pure water they also survived, although with lower CFU counts (Fig. 1b). Clearly, *B. subtilis* cells can survive prolonged nutrient starvation conditions without having to resort to sporulation.

To confirm that non-sporulating *B. subtilis* cells are also present in wild-type populations under deep starvation conditions, the number of sporulating and non-sporulating cells was monitored in a wild-type *B. subtilis* culture. As shown in Fig. 1c, eventually most cells sporulated, however, a substantial fraction of the population (~20–30%) comprised non-sporulating cells, even after more than 100 days of deep starvation. For the first 14 days of starvation, this was confirmed by counting spores and non-sporulating cells by phase contrast microscopy (Supplementary Figure 2B). Comparable results were found when a wild-type culture was resuspended into pure water (Supplementary Figure 2C).

*B. subtilis* spores can germinate spontaneously even when conditions are unfavorable for growth. It is assumed that by this mechanism spores can scout the environment[16]. To examine whether there is a germination-sporulation cycle during long starvation conditions, we measured the viable count of a mutant that is unable to germinate[17]. During the first 11 days of starvation, this strain followed the same reduction in viable counts as the non-sporulating Δ*spoIIE* strain. After that the viable count reduced approximately one more log-unit and stabilized around $10^4$ CFU ml$^{-1}$ (Supplementary Figure 3). The extra reduction in viable counts after 11 days suggests that a fraction of the germination mutant formed spores. However, this reduction did not continue, at least for another 40 days (Supplementary Figure 3), indicating that *B. subtilis* does not use a continuous sporulation-germination cycle to survive long-term starvation. Thus, in contrast to the general assumption, *B. subtilis* does not necessarily have to sporulate to survive long periods of nutrient starvation. For reasons of practicality, the rest of the starvation experiments were performed with the non-sporulating Δ*spoIIE* strain, and CFU measurements were limited to 14 days' incubation in starvation buffer.

**Morphological changes**. Morphological changes caused by long-term nutrient deprivation, primarily a reduction in cell size, have been observed for several bacterial species[2,5,6]. When a *B. subtilis* Δ*spoIIE* culture was filtered and incubated in starvation buffer, we noticed a 40% reduction in cell length over the first 2 days, and cells became almost coccoid (Fig. 2a, b). It should be mentioned that the time axis in Fig. 2b depicts the days of incubation in starvation buffer after filtration. Consequently, at time point day 0 cells have already undergone 2 days of stationary phase in the SMM.

**Antibiotic sensitivity**. The reduction in cell length in the first 2 days of deep starvation suggested that the starved cells are still able to divide. After the cells became so short that they reached an almost coccoid shape, the cell length remained constant (Fig. 2b). This raised the question how metabolically active these small

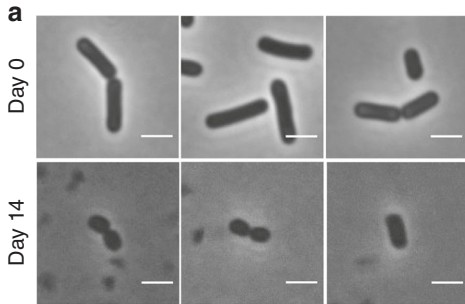

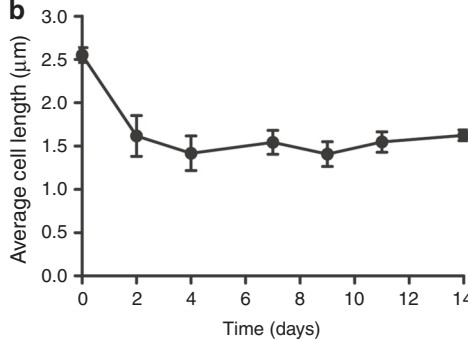

**Fig. 2** Cell length reduction during deep starvation. **a** Phase contrast images of *B. subtilis* Δ*spoIIE* cells in the beginning (day 0), and after 14 days of incubation in starvation buffer. Scale bar is 2 μm. **b** Average cell length. Average and standard deviation from three independent experiments are depicted. Approximately 100 cells were measured for every condition

non-sporulating cells are when they have been incubated for up to 2 weeks in buffer lacking any nutrients. Previous studies used antibiotic sensitivity as an indication of metabolic activity in starved cells[6,18]. The addition of chloramphenicol to starved non-sporulating *B. subtilis* cells had no effect on survival (Fig. 3a). Ampicillin efficiently killed exponentially growing cells, but nutrient-starved cells were much more tolerant to this bactericidal antibiotic (Fig. 3b). A starvation period of 7 and 14 days also resulted in an increased tolerance to paraquat, an inducer of oxidative stress[19] (Fig. 3c).

These data suggested that the small coccoid-shaped *B. subtilis* cells are dormant. To test whether these cells still maintain a membrane potential, cells were treated with the membrane potential-dissipating ionophore valinomycin. This led to a clear decrease in CFUs (Fig. 3d). When we measured the membrane potential in individual cells using the voltage-sensitive dye 3,3′-dipropylthiadicarbocyanine iodide (DiSC$_3$(5))[20], deep-starved cells appeared to have a membrane potential levels almost comparable to those found in exponentially growing cells (Fig. 3e).

**Protein synthesis capacity**. The presence of a membrane potential strongly suggests that the deep-starved cells are still metabolically active. To test whether these cells are capable of expressing proteins, an isopropyl β-D-1-thiogalactopyranoside (IPTG)-inducible green fluorescent protein (GFP) reporter fusion was introduced into the cells. Samples of 0-, 7-, and 14-day-starved cultures were taken and incubated with 1 mM IPTG for 4 h after which GFP levels were measured using fluorescence light microscopy. Surprisingly, 14-day-starved cells were expressing GFP, and in fact better than day 0 cells (Fig. 4). Apparently, the protein synthetic machinery is better adapted to starvation conditions after several days than at the onset of starvation.

To confirm that starved cells were metabolically active, we determined nascent peptide synthesis in individual cells by incubation with the amino-acid analog L-homopropargylglycine (L-HPG) for 18 h followed by fluorescent labeling with Alexa594 using click chemistry[21]. Alexa594 levels in individual cells was measured using fluorescence light microscopy and showed a clear accumulation of the fluorophore in 14- and 19-day-starved cells further incubated with L-HPG (Supplementary Figure 6A, B). Heat-inactivation (15 min 80 °C) or the addition of 200 µg ml$^{-1}$ puromycin completely blocked the incorporation of L-HPG. Finally, we labeled nascent peptidoglycan using the fluorescent D-amino acids analog NBD-amino-D-alanine (NADA)[22], and found that the cell wall of starved cells showed a clear fluorescence after 24 h incubation with NADA (Supplementary Figure 6C). Together, these data show that the surviving cells are still metabolically active despite 2 weeks of starvation.

**Transcriptome analysis**. The small coccoid *B. subtilis* cells and classic stationary phase *B. subtilis* cells have in common that they show tolerance to antibiotics (e.g. Fig. 3, day 14 and day 0, respectively). This is not surprising since both types of cells are dealing with nutrient-limiting conditions that suppress growth. On the other hand, the small coccoid *B. subtilis* cells are capable of synthesizing high levels of protein (Fig. 4). To examine whether deep-starved *B. subtilis* cells differ from stationary phase cells, we performed an RNA-sequencing experiment to compare the transcriptome profiles of both cell types. To obtain sufficient cells after 14 days, a 1 l culture was used as starting material. For the comparison with stationary growth phase cells, a culture was grown in SMM and incubated overnight. The transcriptome profile of an exponentially growing culture was also included in the analysis. As shown in Fig. 5a, a principal component analysis indicated that the 14-day-starved cells differ substantially from both stationary phase and exponentially growing cells. This is also illustrated by a heat map of the transcriptome profiles (Fig. 5b). One hundred-sixty-five genes were at least twofold upregulated in the 14-day-starved culture when compared to the exponentially culture, and 145 genes were at least twofold upregulated in the 14-day-starved cultures when compared to the stationary phase cultures (Fig. 5c). Thirty of these genes were only upregulated in the 14-day-old cells (Table 1). The majority of these genes are involved in nutrient acquisition (*araE*, *exuT*, *rbsC*, *rbsD*, and *rhiF*) and utilization (*acoABC*, *araBDLNPQ*, *ganB*, *hutH*, *manA*, *uxaC*, and *uxuA*). Twenty-one genes were uniquely downregulated in the 14-day-old cells compared to both the exponential and stationary phase cells (Fig. 5c). The majority of these genes are involved in biofilm formation (*bioI*, *epsD*, *epsI*, *epsJ*, *epsL*, and *tasA*) or competence development (*comC*, *comGC*, *comGD*, *comGE*, *comGF*, and *comGG*) (Table 1).

**Growth and cell division**. The transcriptome data indicated that the 14-day-starved cells differed substantially from stationary phase cells, and the DiSC₃(5) assay (membrane potential) and GFP induction experiments suggested that 14-day-starved cells are metabolically active. Also cell division genes, such as *ftsZ*, *ftsA*, *ftsL*, *pbpB*, and *divIVA*, were not downregulated in the 14-day-old culture compared to the exponentially growing culture. This forced us to consider the possibility that these 14-day-starved coccoid *B. subtilis* cells are not dormant but are actually growing. As a first test we examined the localization of the cell division protein DivIVA in deep-starved cells using a DivIVA-GFP reporter fusion. DivIVA is involved in cell division regulation and accumulates at midcell when synthesis of the division septum creates a strong concave (inward) curvature of the cell membrane to which DivIVA specifically binds[23,24]. Microscopic images of

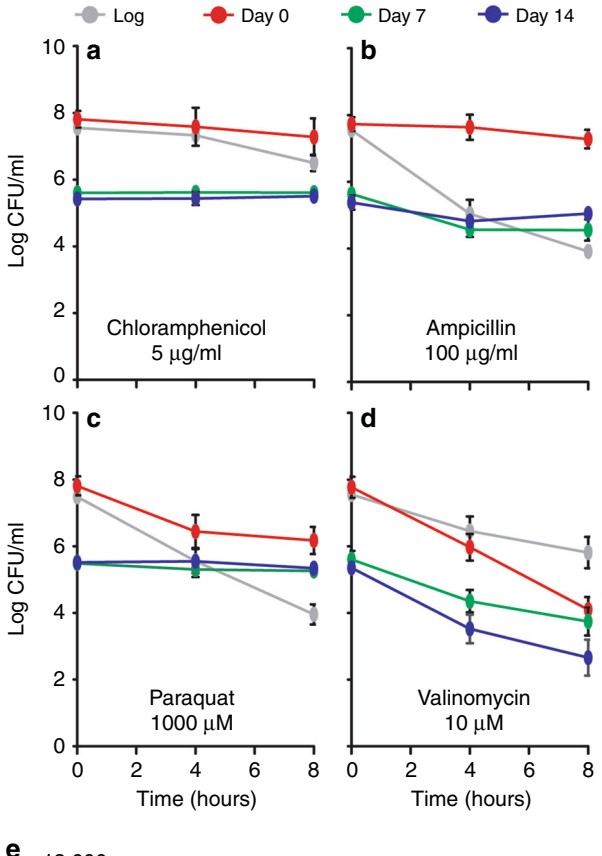

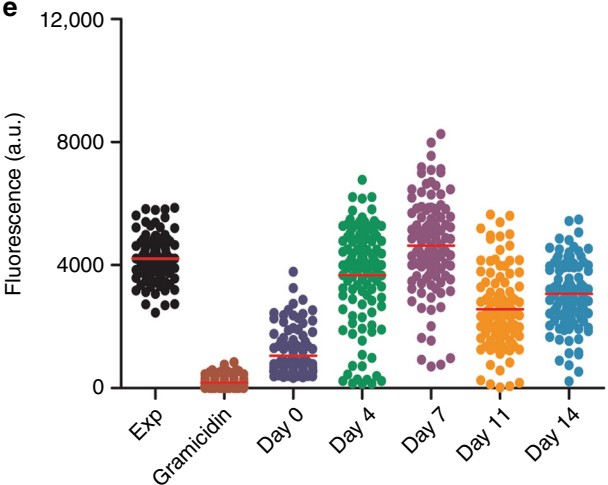

**Fig. 3** Antibiotic sensitivity under starvation conditions. Cultures of ΔspoIIE were set up in accordance with the starvation assay. Samples were taken from cultures at days 0, 7, and 14, and treated with either: **a** 5 µg ml$^{-1}$ chloramphenicol, **b** 100 µg ml$^{-1}$ ampicillin, **c** 1 mM paraquat, or **d** 10 µM valinomycin, for 8 h. As a control, exponential phase cells were treated in the same manner as the starvation samples (gray lines). Graphs show the average and standard deviation of three independent experiments. **e** Membrane potential levels of cells after 0, 4, 7, 11, and 14 days' deep starvation. Relative membrane potential levels were determined based on the uptake of the membrane potential sensitive fluorescent dye 3,3'-dipropylthiadicarbocyanine iodide. As controls, membrane potential levels of exponentially growing cells (OD₆₀₀ 0.2), and cells treated with the ionophore gramicidin ABC (10 µg ml$^{-1}$) were determined. Fluorescence intensities of approximately 100 cells were quantified (median in red). Two biological replicates are presented in Supplementary Figure 4

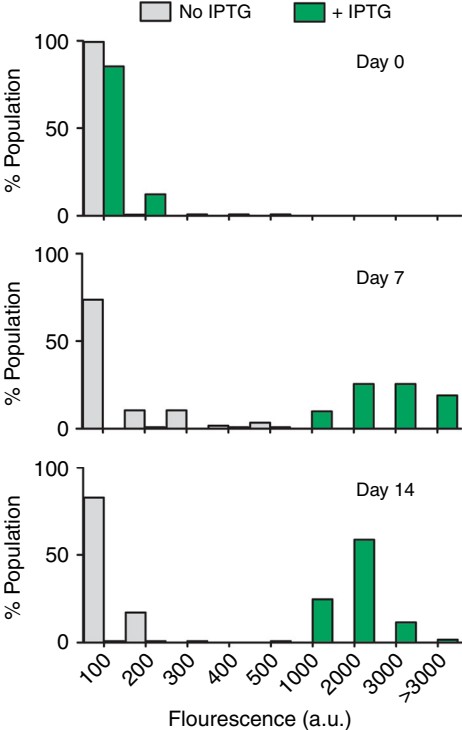

**Fig. 4** *gfp* expression capacity during deep starvation. *B. subtilis* strain Δ*spoIIE* containing an inducible green fluorescent protein (GFP) reporter (strain DG017, *amyE::Phyperspank-sfGFP*) was incubated in starvation buffer for 0, 7, and 14 days, followed by incubation in the presence or absence of 1 mM isopropyl β-ᴅ-1-thiogalactopyranoside (IPTG) for 4 h, after which cells were imaged by fluorescence light microscopy. The fluorescence of approximately 100 cells was quantified for each condition. Cells were binned based on their fluorescence intensities (arbitrary units), and the number of cells in each bin were plotted as histograms. Two biological replicates can be found in Supplementary Figure 5

cells that had undergone 14 days of deep starvation showed a clear fluorescence signal at midcell, suggesting that these cells were still actively dividing (Fig. 6a). When we counted these cells, the frequency of midcell localized DivIVA-GFP seemed to increase during the first 7 days of deep starvation (Fig. 6b).

The presence of cell division proteins at midcell is no definitive proof for growth. To unequivocally determine whether long-term starvation resulted in dormant or in growing cells, we developed a new assay based on blocking cell division. The compound 3-methoxybenzamide (3-MBA) impairs polymerization of the key cell division protein FtsZ of *B. subtilis*, resulting in inhibition of cell division and consequently cell elongation[25]. In theory, this should reveal whether starved cells are either dormant or still growing. Samples were taken over a 14-day starvation period at regular time intervals and cells were incubated in the presence or absence of 3-MBA for 48 h. Cells were stained with the fluorescent membrane dye FM5-95 to facilitate cell length measurements (Fig. 7a). Consistent with the previous cell length measurements (Fig. 2), a reduction in cell length was observed for untreated day 0 samples after 48 h. As expected, this reduction was suppressed when cell division was blocked by 3-MBA. Interestingly, after 2 days of starvation the presence of 3-MBA resulted in a clear increase in cell length, and this was even the case for cells that were starved for 14 days (Fig. 7a, b). We also tested starved wild-type cells and found a similar result (Supplementary Figure 7A). 4′,6-diamidino-2-phenylindole staining of 3-MBA-treated cells showed that these elongated cells contained normal nucleoids (Supplementary Figure 7B), indicating that cell elongation is accompanied by chromosome replication. Importantly, when we tested an *ftsZ* mutant (G196S, N263K) that is resistant to 3-MBA[26], no increase in cell length was observed (Supplementary Figure 7C). We must therefore conclude that deep starvation does not lead to dormant cells but to actively growing cells.

With an average increase in cell length of 0.75 μm after 48 h incubation with 3-MBA (Fig. 7b), and an average cell length of ~1.5 μm (Fig. 2), the doubling time of these deep-starved *B.*

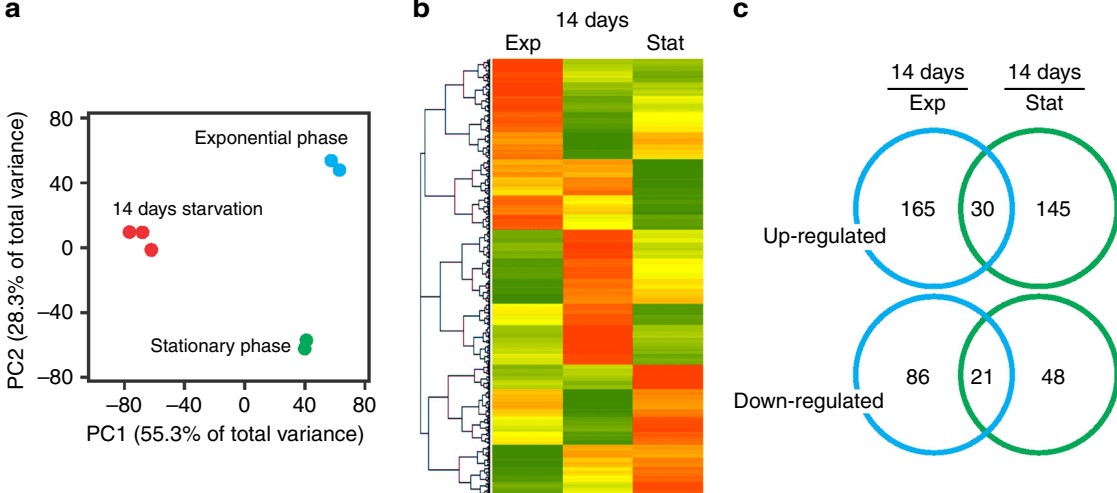

**Fig. 5** Transcriptome comparison. Transcriptomes (RNA-seq) of *B. subtilis* Δ*spoIIE* were analyzed for cells incubated 14 days in starvation buffer (in triplicate), and for cells grown in Spizizen minimal medium (SMM) and harvested in exponential or stationary growth phase, respectively (each in duplicate). **a** Principal component analysis plot of the different samples. Samples were plotted against the first two principal components calculated from the gene expression values. The axes labels indicate percentage of total variance that is explained by each component. **b** Heat map of the averaged transcriptome profiles. Expression levels were transformed to Z-scores. Green indicates low expression (Z-score = −3.5), yellow indicates average expression (Z-score = 0), and red indicates high expression (Z-score = +3). **c** Venn diagram indicating the number of upregulated and downregulated genes in 14-day-starved cells compared to exponential growth phase cells (left) and stationary growth phase cells (right). Strain used: *B. subtilis* PG344 (*spoIIE::erm*)

**Table 1 Up- and downregulated genes in 14-day-starved cells**

| Gene | Function | Fold upregulation 14 days/exp | Fold upregulation 14 days/stat |
|---|---|---|---|
| abnA | Arabinan degradation | 3.1 | 2.6 |
| acoA, acoB, acoC | Acetoin utilization | 8.5, 4.8, 2.0 | 12.7, 3.4, 2.3 |
| araB, araD, araL, araN, araP, araQ | Arabinose utilization | 2.3, 2.6, 2.4, 2.1, 3.6, 3.6 | 2.2, 2.0, 2.2, 2.3, 2.1, 2.5 |
| araE | Uptake of arabinose, galactose, and xylose | 2.1 | 2.0 |
| cwlT | Conjugative transfer of ICEBs1 | 2.5 | 2.2 |
| czcD | Cation efflux, resistance against Zn, Cu, Co, Ni | 2.6 | 2.7 |
| exuT | Hexuronate uptake | 2.1 | 3.1 |
| ganB | Galactan utilization | 2.9 | 3.4 |
| hutH | Histidine utilization | 2.4 | 2.2 |
| manA | Mannose utilization | 2.7 | 2.2 |
| mmgB | Mother cell metabolism | 32.5 | 4.4 |
| nicK | Conjugation of ICE BS1 | 4.7 | 3.9 |
| rbsC, rbsD | Ribose uptake | 3.5, 3.1 | 2.1, 2.3 |
| rhiF | Uptake of rhamnose oligosaccharides | 2.5 | 4.3 |
| sigV | Resistance to lytic enzymes | 2.8 | 3.6 |
| uxaC, uxuA | Hexuronate utilization | 2.5, 2.0 | 3.0, 2.1 |
| ydfQ | Unknown | 2.2 | 3.5 |
| ydfR | Unknown | 19.7 | 2.7 |
| ydzR | Unknown | 4.3 | 3.6 |
| yezD | Unknown | 3.4 | 4.1 |
| yisJ | Unknown | 17.2 | 3.0 |
| | | **Fold downregulation 14 days/exp** | **Fold downregulation 14 days/stat** |
| biol | Biosynthesis of biotin | −2.3 | −2.1 |
| comC | Genetic competence | −2.4 | −3.0 |
| comGC, comGD, comGE, comGF, comGG | Genetic competence | −3.1, −3.3, −2.8, −3.1, −3.4 | −3.5, −3.5, −2.9, −3.0, −2.4 |
| coxA | Resistance of the spore | −2.7 | −2.1 |
| epsD, epsI, epsJ, epsL | Biofilm formation | −4.6, −8.1, −4.6, 8.4 | −2.8, −4.5, −3.1, −5.0 |
| nadA, nadC | NAD biosynthesis | −2.1, −2.4 | −2.0, −2.3 |
| rtpA | Regulation of tryptophan biosynthesis | −4.2 | −2.6 |
| tasA | Biofilm formation | −5.1 | −2.7 |
| thiS | Biosynthesis of thiamine | −3.5 | −3.5 |
| thiU | Thiamine uptake | −2.2 | −2.2 |
| yckD | Unknown | −3.9 | −2.4 |
| ywcJ | Unknown | −2.0 | −2.3 |
| ywpE | Unknown | −23.6 | −8.3 |

The transcriptome profile of 14-day-starved cells was compared to those of exponential phase and stationary phase cells. Genes that showed at least a twofold up- or downregulation in both situations, with p-values lower than 0.05, are listed. Genes in an operon have been grouped together

subtilis cells is estimated to be approximately 4 days. Since these cells show such a distinct transcriptome profile and cell shape, and since they are still growing, we propose to call this the "oligotrophic growth state".

**Nutrients**. Oligotrophic growth implies the presence of small levels of nutrients in the starvation buffer. A likely source is the cells that lyse in the first days of incubation. To confirm this, 7-day-old cells were filtered and resuspended either in the filtrate or in fresh starvation buffer. As shown in Fig. 8a, resuspension in starvation buffer resulted in a further decline in the CFU, whereas resuspension in the culture filtrate had no effect on the CFU count. Clearly, cell lysis is necessary to provide nutrients to maintain the oligotrophic growth state of the surviving cells. This also explains why stationary phase cells, when filtered and resuspended into fresh starvation buffer (day 0 samples), displayed a lower membrane potential and a low capacity to express GFP than later time samples (Figs. 3e and 4).

A Bradford assay to quantify protein concentrations indicated that the amount of proteins released from lysed cells reached approximately $120 \, \mu g \, ml^{-1}$ after 4 days of starvation (Supplementary Figure 8). To confirm that this low concentration of proteins is indeed sufficient to support oligotrophic growth,

7-day-starved cultures were filtered and resuspended in starvation buffer containing low concentrations of either tryptone or peptone. It appeared that even concentrations as low as $0.5 \, \mu g \, ml^{-1}$ tryptone or peptone were sufficient to maintain the oligotrophic growth state (Fig. 8b).

**Genetic factors**. To determine whether the oligotrophic growth state requires specific gene products, several of the upregulated genes in Table 1 were deleted (acoA, manA, mmgB, yezD, and yisJ) and tested for survival under deep starvation conditions. However, none of the five tested deletions reduced the viability (Supplementary Figure 9A).

Several of the upregulated genes in Table 1 are controlled by alternative sigma factors[27,28]. We first tested whether one of the sporulation sigma factors (ΔsigE, -F, -G, -H, and -K) was required for survival, but none of the sigma factor deletion mutants showed an effect (Supplementary Figure 9B). Also mutants lacking either the general stress response sigma factor SigB, the heat shock sigma factors SigI, or the sigma factor SigL, which renders cells cold-sensitive, were able to survive for 14 days without problems (Supplementary Figure 9C). Last, we tested the extra-cytoplasmic function (ECF) sigma factors (SigM, -V, -W, and -X) involved in cell envelope stress[29]. As shown in Fig. 8c, the

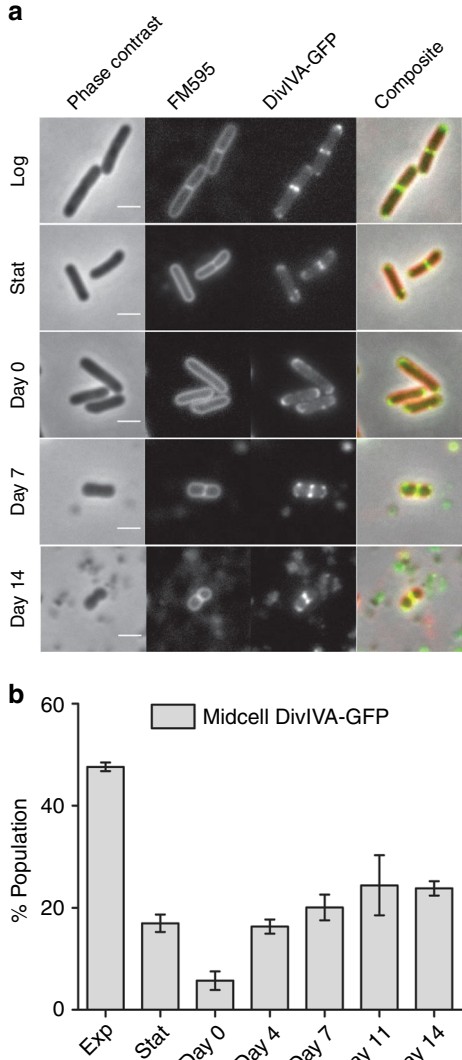

**Fig. 6** Presence of cell division sites. *B. subtilis* Δ*spoIIE* cells encoding *divIVA-gfp* reporter were incubated for 14 days in starvation buffer. **a** At different time intervals samples were taken for microscopic analysis. Cells were stained with the fluorescent membrane dye FM5–95. Scale bar is 2 μm. **b** To avoid bias, the captured phase contrast images were used to select approximately 100 cells per sample, and the number of cells containing a midcell DivIVA-GFP signal, indicative of cell division, were counted. As a control, the number of division sites present in both exponentially growing cells ($OD_{600}$ ~0.2), and stationary growth phase cells (overnight) were determined. Bar diagram represents average and standard deviation of three independent experiments. Strain used: *B. subtilis* DG001 (*spoIIE::erm, divIVA:divIVA-GFP(cm)*)

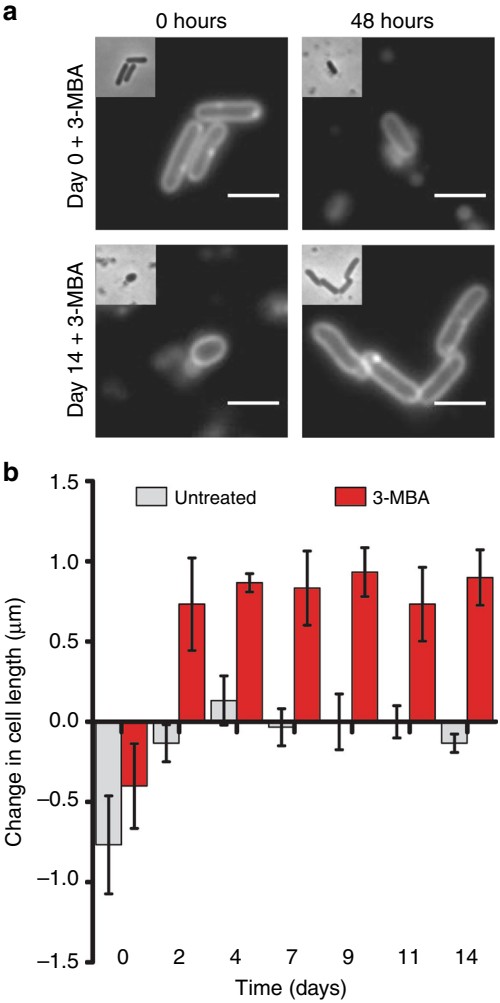

**Fig. 7** Cell growth under deep starvation conditions. *B. subtilis* Δ*spoIIE* cells (strain DG001) were incubated for 14 days in starvation buffer. At regular time intervals samples were withdrawn and incubated with the cell division inhibitor 3-methoxybenzamide (3-MBA) for 48 h. **a** Cells were stained with the fluorescent membrane dye FM5-95 in order to determine cell length before and after 3-BMA treatment. Scale bar is 2 μm. **b** Average change in cell length was calculated for approximately 100 individual cells for each time point. Bar diagram depicts the average and standard deviation of three independent experiments

Δ*sigX* mutant showed a continuous decline in viable count, reaching an order of magnitude lower CFU counts after 14 days, suggesting that the oligotrophic cells are exposed to cell wall disturbing factors. To examine whether bacteriophages are responsible for this, we tested a strain lacking all six prophages[30]. However, this strain showed a reduction in CFUs that was comparable to the wild type (Supplementary Figure 9D). A strain lacking the major autolysin LytC, involved in cell separation and cell wall turnover, also had no strong effect on survival (Supplementary Figure 9D). We then examined whether the main stress proteins are necessary for survival and tested strains lacking either the protein quality control protease ClpP, the

stringent response regulator RelA, or the DNA recombination/repair proteins RecA and PnpA[31,32]. Only the Δ*relA* mutant showed a slightly lower viability after 7 and 14 days of deep starvation (Supplementary Figure 9D).

Differentiation processes in *B. subtilis* are coordinated by a few key transcription factors. Motility is activated by the sigma factor SigD, genetic competence by the activator ComK, and transcription factors like SinI, CodY, AbrB, DegS, DegU, and Spo0A regulate differentiation processes such as sporulation, degradative enzyme production, and biofilm formation[10,33,34]. However, none of these transcription factors were essential to survive deep starvation, although the absence of DegU caused a reduction in viable count, especially during the first week of starvation (Fig. 8d). The response regulator DegU controls expression of secreted proteases and it is likely that proteases are necessary for the release of nutrients from lysed cells. Indeed, a strain lacking all eight extracellular secreted proteases (strain ΔWB800 lacking *nprB*, *aprE*, *epr*, *bpr*, *nprE*, *mpr*, *vpr*, and *wprA*)[35]

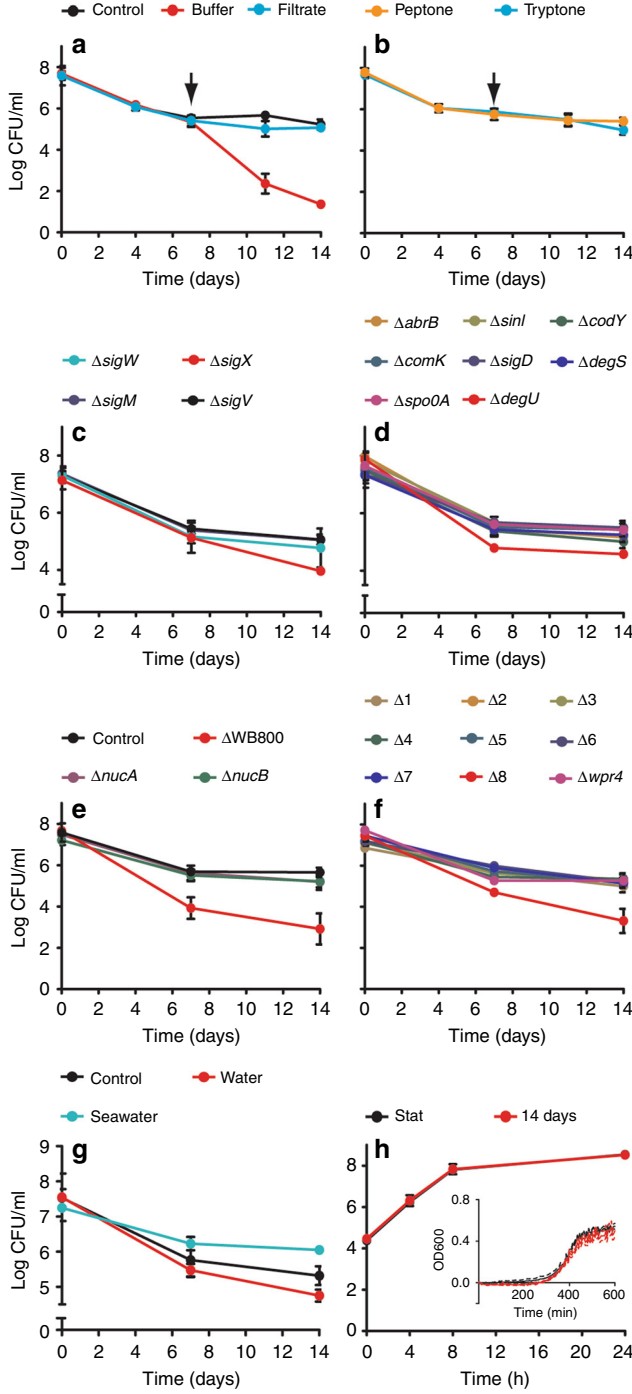

**Fig. 8** Nutrient and genetic factors, and escape. **a** Importance of nutrients released by lysed cells was determined by filtering *B. subtilis* Δ*spoIIE* cultures (strain DG001) after 7-day starvation followed by resuspension (arrow) in either fresh starvation buffer (buffer) or the cell filtrate (filtrate). **b** Effect of resuspension, after 7-day starvation, in starvation buffer supplemented with 0.5 μg ml$^{-1}$ peptone or tryptone. **c** Survival of different extra-cytoplasmic function (ECF) sigma factor mutants. Only log 4–8 colony-forming units (CFU) ml$^{-1}$ is shown to emphasize differences. **d** Survival of mutants lacking different stationary phase regulators. Only log 4–8 CFU ml$^{-1}$ is shown to emphasize differences. **e** To determine whether proteases are important for survival, a strain lacking all eight secreted proteases, ΔWB800, was tested. The absence of the two extracellular nucleases produced by *B. subtilis* on survivability was also tested (Δ*nucA* and Δ*nucB*). **f** Survival of strains carrying cumulative deletions of the eight secreted proteases (*nprB, aprE, epr, bpr, nprE, mpr, vpr,* and *wprA,* respectively). *wprA* was the last deleted gene (Δ8) but a single *wprA* deletion (Δ*wprA*) shows normal CFU levels. **g** Survival of oligotrophically growing cells in seawater. Cells were incubated for 14 days in starvation buffer, water, or artificial seawater (450 mM NaCl, 10 mM KCl, 9 mM CaCl$_2$, 30 mM MgCl$_2$, and 30 mM MgSO$_4$). Only log 5–9 CFU ml$^{-1}$ is shown to emphasize differences. **h** Escape from the oligotrophic growth state. SMM was inoculated with either stationary phase cells or 14-day-starved cells, and outgrowth was measure by either CFU or optical density (OD) measurements (inset). Graphs represent averages and standard deviations of three independent experiments. The Δ*spoIIE* background strain was used in all experiments

soil[36]. In fact, *B. subtilis* is not only found in soil but can also be isolated from seawater[37]. When we used artificial seawater for deep starvation experiments, the viability increased, and CFU counts became 10× higher, indicating that *B. subtilis* can maintain oligotrophic growth in salty seawater (Fig. 8g).

Finally, we tested how much time it takes for cells to escape the oligotrophic growth state and resume normal growth. To this end, samples from 14-day-starved cultures were inoculated in fresh SMM medium. For comparison, normal stationary phase cells were used that were grown overnight in SMM medium. The overnight culture was diluted 10$^3$ times to obtain comparable viable counts at $t = 0$. Growth was followed by either viable count measurements or optical density in microtiter plates. As shown in Fig. 8h, the 14-day-starved cells resumed growth as efficiently as cells from the overnight culture. Thus, cells in the oligotrophic growth state are capable of rapidly responding to improved nutrients conditions.

## Discussion

A recurring question is whether cells become dormant when exposed to long-term starvation conditions or whether they are still metabolically active. The main reason for this is that an unambiguous assay for (very slow) cell growth has been lacking. One of the longest microscopic observations, 1.5 days monitoring of starved *E. coli* cells in a microfluidic setup, did not detect cell growth[38]. On the other hand, in a long (weeks) starvation experiment using *E. coli*, mutants appeared that could outcompete non-starved cells, suggesting that starved cells were still active[39]. However, it should be noted that in this experiment the starved cells were kept in the same growth medium (lysogeny broth (LB) medium) and were not resuspended in fresh water or buffer lacking any nutrients. Other long-term starvation studies, e.g. with *M. luteus, Deinococcus sp.,* and *E. coli,* concluded that deep starvation resulted in dormant cells[5,40,41]. Prolonged nutrient starvation can also result in viable-but-non-culturable (VBNC) cells[42–44]. These VBNC cells appear metabolically active when observed with certain fluorescent probes, but they can no

showed a 100-fold decrease in survival (Fig. 8e). To determine whether this was due to the absence of a specific protease, we introduced sequential deletions of the eight protease genes into the *spoIIE* background. As shown in Fig. 8f, only the strain lacking all eight proteases showed a decrease in viable count. *wprA* was the last deleted gene (Δ8) but a single *wprA* deletion (Δ*wprA*) showed normal CFU levels (Fig. 8f). This demonstrated that a single secreted protease is sufficient for oligotrophic growth. We also tested whether the absence of the extracellular nucleases NucA or NucB made a difference, but this was not the case (Fig. 8e).

**Salt resistance and escape.** Oligotrophic environments are ubiquitous and include most lakes and seas, but can also be found in

longer grow when nutrients are replenished[45]. In our experiments the drop in CFU appears to be caused primarily by cell lysis rather than the appearance of VBNC cells, since the optical density of starved cultures decreased substantially and microscopic observations also showed a strong reduction in cell numbers.

Here we have shown that deep starvation of *B. subtilis* results in actively growing cells, however this growth does not lead to an increase in CFUs. This balance between cell growth and cell lysis has been coined cryptic growth[46]. This phenomenon is presumably widespread in nutrient-depleted cultures but it is difficult to measure, and previously could only be assessed indirectly by testing sensitivity toward antibiotics or the incorporation of radioactive precursors. Our cell division inhibition assay with 3-MBA provides now an unambiguous test for cryptic growth, enabling direct detection of slow growing cells at a population-wide scale.

A fascinating question is how cell growth and cell lysis are balanced during cryptic growth. In *B. subtilis*, the matured endospore escapes the mother cell by programmed cell lysis[47]. In addition, *B. subtilis* cells can display a cannibalism phenotype during stationary phase, releasing killing factors that specifically lyse other *B. subtilis* cells, thereby releasing nutrients into the medium[48]. Whether these mechanisms are involved in cryptic growth-related cell lysis remains to be determined. However, both cell lysing mechanisms are activated by Spo0A, and since a Δ*spo0A* mutant shows the same lysis pattern as wild-type cells, other mechanisms might be active. We can rule out the involvement of bacteriophages, given that a strain devoid of all bacteriophages showed a normal oligotrophic growth pattern (Supplementary Figure 9D). It has been shown that depletion of the proton motive force causes autolysis of *B. subtilis* cells[49]. After 14 days of deep starvation, the membrane potential is almost comparable to that of normally growing cells in many cells. However, there is a large heterogeneity in membrane potential levels (Fig. 3e), and it is possible that cells with low membrane potentials are the ones that are prone to lysis.

The oligotrophic *B. subtilis* cells showed an increased tolerance to chloramphenicol, ampicillin, and oxidative stress. Tolerance to environmental stresses during nutrient deprivation is well-known. For example, starved *E. coli* and *Vibrio* cells have been shown to become tolerant to hydrogen peroxide, heat, and ultraviolet (UV) radiation[50,51]. Starved *S. typhimurium* cells develop tolerance to UV radiation and hydrogen peroxide, as well as tolerance to acids[52]. Tolerance to antibiotics by reduction of the growth rate has been put forward as the main reason for the bacterial persister phenotype[53]. During oligotrophic growth of *B. subtilis* the doubling time is reduced more than 100× compared to normal exponentially growing cells (from ~40 min to ~4 days). This dramatic reduction in growth rate might explain why oligotrophic cells are tolerant to chloramphenicol and ampicillin and cope better with oxidative stress. This property might also explain why mutants with either reduced DNA repair (Δ*recA*), cold shock survival (Δ*pnpA*), protein refolding (Δ*clpP*), or general stress survival (Δ*sigB*) capacities did not show a decrease in viability during the 14 days' starvation.

Whether oligotrophic growth is a carefully regulated differentiation process remains to be seen. Of the transcriptional regulators tested in this study, only the absence of SigX, DegU, and RelA showed an effect on viable counts during starvation. However, the absence of the stringent response regulator RelA, and the two-component response regulator DegU involved in the induction of secreted proteases, had only mild effects on survival. The strongest effect, one order of magnitude reduction in viable count after 14 days, was observed for the Δ*sigX* mutant. The ECF sigma factor SigX induces genes that modify the cell wall, making

cells more resistant to heat and certain cell envelope-targeting compounds[54,55].

Oligotrophic cells have to be thrifty and the turnover of proteins will be slow, as the tolerance against chloramphenicol indicated. In addition, the extreme reduction in growth rate means that for most processes only a limited number of proteins are required to fulfill the necessary tasks. This is for example illustrated by the fact that only a single protease is required to maintain oligotrophic growth (Fig. 8f). Transcription is an intrinsically noisy process[56] and some leaky expression will occur especially over long time intervals. Therefore, a carefully regulated signal transduction pathway might not be essential in these slowly growing cells, and substrate feedback at the enzyme level might be the predominant measure of regulation during oligotrophic growth.

Some bacteria are obligate oligotrophs whereas others are facultative oligotrophs[57]. *B. subtilis* can now be added to the latter family. We have shown that concentrations as low as 0.5 μg ml⁻¹ tryptone or peptone are sufficient to maintain oligotrophic growth. Normally, *B. subtilis* is studied using rich growth media such as LB. LB is composed of 10 mg ml⁻¹ tryptone and 5 mg ml⁻¹ yeast extract, thus, concentrations that are approximately 10,000 times higher than necessary for oligotrophic growth. Since *B. subtilis* lives in oligotrophic environments such as soil and seawater, the question arises whether growth in rich media like LB is the most natural condition to investigate this organism, and whether oligotrophic growth might not be the most common cellular state of this organism in nature.

It will be interesting to see whether oligotrophic growth is a common survival strategy for bacteria. A large drop in CFU during the first days of deep starvation is observed for most bacteria that are starved for prolonged periods of time, including *S. aureus*, *E. coli*, *M. luteus*, *S. typhimurium*, *B. suis*, and *F. columnare*[2,3,6,58,59]. Consequently, at least the small amount of nutrients necessary to support oligotrophic growth will be available.

Thus far, sporulation has been considered the principal cellular differentiation through which *B. subtilis* survives prolonged periods of nutrient depletion. Here we show that *B. subtilis* can also survive by growing very slowly, a state that makes them resilient to a variety of environmental stresses. The advantage of oligotrophic growth over sporulation is that oligotrophic cells are able to rapidly resume normal growth when nutrient conditions improve (Fig. 8h), whereas spore germination is a time-consuming and extremely heterogenic process, with many spores delaying outgrowth even when conditions are good[16,60]. In conclusion, oligotrophic growth adds another differentiation state to the already rich repertoire of cellular differentiations that *B. subtilis* is capable of.

## Methods

**Maintenance and growth of strains.** Nutrient agar (NA) (Oxoid) was used for routine selection and maintenance of both *B. subtilis* and *E. coli* strains. Supplements were added as required: chloramphenicol (cm, 5 and 50 μg ml⁻¹), erythromycin (erm, 1 μg ml⁻¹), kanamycin (kan, 2 μg ml⁻¹), spectinomycin (spec, 50 μg ml⁻¹), tetracycline (tet, 10 μg ml⁻¹), ampicillin (amp, 100 and 1000 μg ml⁻¹), valinomycin (val, 10 and 100 μM), IPTG (1 mM), and paraquat (para, 100 and 1000 μM). For liquid cultures, cells were cultured in either SMM[61] (15 mM (NH₄)₂SO₄, 80 mM K₂HPO₄, 44 mM KH₂PO₄, 3 mM tri-sodium citrate, 0.5% glucose, 6 mM MgSO₄, 0.2 mg ml⁻¹ tryptophan, 0.02% casamino acids, and 0.00011% ferric ammonium citrate (NH₄)₅Fe(C₆H₄O₇)₂), LB (10 g l⁻¹ tryptone, 5 g l⁻¹ peptone, and 10 g l⁻¹ NaCl), or Penassay broth.

**Strains.** Strains and primers used in this study are listed in Supplementary Table 1 and 2, respectively. Strain construction was achieved through transformation of recipient strain with chromosomal DNA from the appropriate strain, and checked by selecting for the correct resistance. For strain DG034, the erythromycin resistance cassette was removed using plasmid pDR244 according to the protocol

provided by the Bacillus Genetic Stock Center. Briefly, pDR244 was transformed into the recipient strain and selected for at 30 °C on spectinomycin. Successful colonies were then streaked onto plain NA plates and incubated at 42 °C overnight. The resultant colonies were then checked for sensitivity to both spectinomycin and erythromycin. The removal of the cassette was confirmed by PCR. All strains containing "clean" gene deletions, no antibiotic resistance marker, were verified by PCR using oligonucleotides listed in Supplementary Table 2. In order to replace the erythromycin resistance cassette in strain PG344, marker-exchange using plasmid pErm::Spec was carried out[62].

**Deep starvation conditions and spore assay**. Bacteria were cultured in SMM at 37 °C with aeration for 48 h. Cells were collected by vacuum filtration using a 47-mm filter membrane with 0.45-μm pores (Thermo Fisher Scientific). This was always performed at 37 °C to prevent temperature shock. Filtered cells were resuspended in starvation buffer (15 mM $(NH_4)_2SO_4$, 80 mM $K_2HPO_4$, 44 mM $KH_2PO_4$, 50 mM NaCl, and 0.8 mM $MgSO_4$) by vortexing. The resuspended cultures were incubated at 37 °C while shaking for 14 days. Periodic sampling was performed to determine viable counts through serial dilutions and plating on NA plates. The spoIIE gene was deleted by insertion of an erythromycin resistance marker. This, and the use of the DivIVA-GFP marker in several cases, enabled us to check for possible contaminations during long starvation periods.

Spore numbers were determined by heating samples for 25 min at 80 °C, which will kill normal cells except for spores. A serial dilution on NA plates was used to determine spore counts.

**Microscopy-based methods**. Microscopic experiments were performed using a Nikon Eclipse Ti (Nikon Plan Fluor ×100/1.30 Oil Ph3 DLL and Plan Apo ×100/1.40 Oil Ph3 objectives) microscope. Images were acquired using Metamorph 6 (Molecular Devices, Inc) and analyzed with ImageJ (National Institutes of Health). For visualization, 0.3 μl samples were immobilized on a thin layer of 1.2% agarose and a glass coverslip (VWR) was placed on top.

Cell lengths were determined by staining cell membranes with 0.4 μg ml$^{-1}$ FM5-95 (Molecular Probes), followed by length measurements using ImageJ. To determine changes in cell length, samples were treated with 10 μM of the FtsZ inhibitor 3-MBA for 48 h at 37 °C. Images were acquired at the beginning of incubation and after 48 h.

GFP levels were determined using a GFP-filter set and quantification using ImageJ. On days 0, 7, and 14 of starvation strain DG017 (amyE::spec Phyperspank-sfGFP) was incubated with and without 1 mM IPTG for 4 h at 37 °C, after which cells were observed with fluorescence light microscopy.

Membrane potential determination was achieved using the voltage-sensitive dye $DiSC_3(5)$, which accumulates in polarized cells[20]. Cellular membrane potential levels were assessed for starved cells by incubating cells with 2 μM $DiSC_3(5)$ for 5 min at 37 °C, followed by microscopy using Cy5-filter sets. As a control, the $DiSC_3(5)$-fluorescence levels were measured for exponentially growing cells, and cells depolarized with 10 μg ml$^{-1}$ gramicidin ABC[20].

To determine nascent peptide synthesis, samples were treated with 1 mM of the amino-acid analog L-HPG (Jena Bioscience) for 18 h at 37 °C. The samples were washed with 1× phosphate-buffered saline and subsequently permeabilized using sequential resuspension in 50 and 100% ethanol. The L-HPG incorporated in the nascent peptides was clicked to the azide group in AF594-azide (Jena Bioscience) using click chemistry[21], using THPTA-based CuAAC Cell Reaction Buffer Kit (Jena Bioscience). Cells from microscopy images were auto-selected and quantified using Coli-Inspector[63]. Statistical analysis between HPG+ and other control samples was performed using unpaired t-test in GraphPad Prism.

To visualize peptidoglycan synthesis, fluorescently labeled D-amino acid NADA was used[22]. Efficient peptidoglycan labeling of B. subtilis cells requires a strain lacking the d,d-carboxypeptidase encoded by dacA[22]. ΔdacA (strain DacAKS15) cultures were set up in accordance with the starvation protocol. After 7 and 14 days of starvation, 400 μl samples were taken and incubated with and without 5 μM NADA for 48 h at 37 °C. From these subcultures 100 μl samples were taken after 1 and >24 h. The cells underwent a centrifugation wash, to remove excess NADA, and resuspended in starvation buffer, after which cells were observed with fluorescence light microscopy.

**RNA-sequencing**. B. subtilis (ΔspoIIE) cultures were set up in accordance with the starvation assay. In order to obtain sufficient cell material, 1 l culture volumes were used. Cells were collected by filtering through 90-mm poly-ethersulfone membrane filters (0.22-μm pores (Millipore)) at 37 °C. For comparison, stationary phase and exponential phase cells were collected. For stationary growth phase samples, 1 l ΔspoIIE cultures were grown overnight (16–18 h) at 37 °C under continuous shaking. For the exponential growth phase samples, 1 l ΔspoIIE cultures were grown to an $OD_{600}$ of ~0.2 at 37 °C under continuous shaking. Also the stationary phase and exponential growing cells were collected by filtration. After harvesting cells were flash-frozen in liquid nitrogen, and stored at −80 °C.

To isolate RNA, the frozen pellets were grounded by using a mortar and pestle before immersion in QIAzol Lysis Reagent (Qiagen). RNA was isolated and purified using the RNeasy kit (Qiagen), including an on-column treatment with the

RNase-free DNase set (Qiagen). The concentration was measured on a NanoDrop ND-2000 (Thermo Scientific) and RNA integrity was examined using the 2200 TapeStation System with Agilent RNA ScreenTapes (Agilent Technologies). ERCCs spike-in mix 1 (Thermo Fisher Scientific) were added to each RNA sample and a ribosomal RNA depletion was performed using the Ribo-Zero rRNA Removal Kit (Gram-Positive Bacteria) (Illunnia). Barcoded RNA libraries were generated according to the manufacturers' protocols using the Ion Total RNA-Seq Kit v2 and the Ion Xpress RNA-Seq barcoding kit (Thermo Fisher Scientific). The size distribution and yield of the barcoded libraries were assessed using the 2200 TapeStation System with Agilent D1000 ScreenTapes (Agilent Technologies). Sequencing was performed on the Ion Proton system using the Ion PI Chips (Thermo Fisher Scientific) according to the manufacturers' protocols.

The quality of the sequencing data was assessed using FASTQC[64] and an in-house pipeline. The reference genome and annotation were downloaded from the NCBI (NC_000964.3), and the sequence reads were mapped onto the genome[65]. The gene expression levels were quantified using HTseq. The data were normalized and analyzed for differential expression using DESeq2[66]. Gene expression data have been deposited at the public repository Gene Expression Omnibus, accession number GSE102140. Variance stabilized normalized data were subjected to principal component analysis and cluster analysis, using R statistical software.

**Protein concentration**. Protein concentration of the starvation medium was determined using a Bradford assay (Bio-Rad) according to the manufacturers' instructions. Samples from starvation cultures were taken through the incubation period, cells were removed by centrifugation (5 min at 8000 rpm, ~14,000 × g) and the supernatant used for the Bradford assay.

**Outgrowth experiment**. Both stationary phase cells and 14-day-old cells (ΔspoIIE) were used to inoculate SMM (1:10 diluted). Stationary phase cells were first diluted 1000-fold in order to have approximately the same starting cell count as 14-day-starved cells. Cultures were incubated at 37 °C with aeration. The CFU was determined by taking periodic samples and serial dilution on NA plates. The $OD_{600}$ was determined using a microtitre plate reader (BMG FLUOstar OPTIMA).

**Reporting summary**. Further information on experimental design is available in the Nature Research Reporting Summary linked to this article.

## Data availability

Gene expression data have been deposited in the public repository Gene Expression Omnibus, accession number GSE102140. Other data supporting the findings of this study are included in the paper and its Supplementary Information files, or available from the corresponding author upon request.

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

## Acknowledgements

We would like to thank S. Syvertsson, J. Kirstein, J. Stülke, R. Daniel, A. Guyet, J. Errington, T. Ewen, S. Ishikawa, and C. Harwood for strains; and K. Turgay for critical reading of the manuscript. Funding for D.A.G. was provided by a Biotechnology and Biological Sciences Research Council (BBSRC) DTP Studentship BB/J014516/1. G.D. was funded by EMBO (ALTF 936–2016) and European Commission MCSA-IF grant 749510. P.G. and H.S. were funded by BBSRC grants BB/I004238/1 and BB/I01327X/1,

respectively, and L.W.H. was partially funded by STW-Vici grant 12128 from the Netherlands Organization for Scientific Research (NWO).

## Author contributions

D.A.G. carried out the experiments. D.A.G. and L.W.H. designed the project, and D.A.G., G.D., P.G., H.S., M.J.J. and L.W.H. analyzed the data and wrote the paper.

## Additional information

**Competing interests:** The authors declare no competing interests.

