## [Peer Review File · Nature Communications]

Reviewers' comments:

Reviewer #1 (Remarks to the Author):

This ms. describes interesting observations of the growth of *B. subtilis* under extremely nutrient poor conditions. Given that "natural" environments are more like these conditions, this work is potentially of broad interest and importance. However, many of the claims made by the authors are not sufficiently supported by the experiments described which greatly reduces my overall enthusiasm. In addition, there are a number of confusing statements about the data and important statistical tests and/or experimental details are missing

1. How can the authors state "after 100 days a large fraction of cells were viable" (line 93), but the data in Fig. 1A show that less than 1 in 10x5 cells made cfus?
2. What is the statistical significance of the differences in Fig. 1B? T60 does not look significant. How many biological replicates were conducted?
3. Under these growth conditions, did they confirm that the pH of medium did not change? Certainly with LB, it needs to be buffered for growth >24h
4. It is not true that sporulation "cannot be reversed once started" (line 62) –sporulating cells are not committed to growth until after asymmetric division (Parker & Errington, 1996) when a *sigF* gene is expressed (Dworkin & Losick 2005). Furthermore, while *spoIIE* mutants are unable to complete sporulation, they can initiate sporulation, and in fact the phenotype of a *spoIIE* null mutant is that it is disporic (Illington & Errington 1991).
5. In their analysis of wt cells (Fig. 1C), they interpret their data as showing that a substantial fraction of the population are non-sporulating cells. This interpretation depends on the assumption that the spore cannot germinate under these conditions. Do we know this? A germination defective mutant might be useful, in this respect. There is quite possibly a dynamic equilibrium between spores and non-spores, which would not be revealed by discontinuous sampling (as opposed to continuous observation).
6. In order to make claims about the metabolic state of these cells, then they must directly assay parameters such as DNA replication, RNA synthesis, or protein synthesis using incorporation of radiolabeled precursor. This will give at least an average level of these processes. To definitively show that most of the cells are active (as opposed to only a subset), they must use fluorescent probes, so that they can image single cells.
7. The GFP experiment (Fig. 4) has too many caveats to be interpreted as a strong indicator of protein synthesis. For example, how do they know that the inability to synthesize GFP is not due to an inability to synthesize the *gfp* mRNA?
8. Their indirect assay for growth (Fig. 7) depends on the ability of the 3-MBA compound to block FtsZ polymerization. Since this assay depends on an inhibitor, they need to perform experiments with the FtsZ mutant that is no longer 3-MBA sensitive to ensure that their action of the compound under their experimental conditions is specific to FtsZ. They also need to explain why they are not using other methods which do not depend on inhibitor (such as the TIMER system of Claudi et al 2014).
9. I do not understand their discussion of noise (line 389-395) – how are they measuring noise in their system – is it intrinsic or extrinsic – the data in Fig.4 are not sufficient to make any specific claims.

Reviewer #2 (Remarks to the Author):

The manuscript presents a very broad and very detailed analysis of the impact of carbon starvation on *B. subtilis* cells (adequate nitrogen, phosphorus and sulfur continued to be provided), with most of the data obtained with non-sporulating mutant strains. This approach makes the results of broader relevance to bacterial species, but does limit the interpretation for *B. subtilis* itself. Nonetheless, the results are intriguing and will undoubtedly surprise the majority of readers.

In several cases, the information provided is insufficient to provide a full understanding of what has happened. For instance, in the paragraph beginning on line 94, the reader is not informed as to whether the optical density of long-term cultures was measured as a potential indicator of cell lysis. Also, the statement that starvation resulted in “release of more nutrients” is not supported by any data provided.

Fig. 1: The legend refers to CFU but doesn't name the medium used. In addition, it would be very helpful to include in parts A and B the OD values at each time point. If the samples in parts A and B were equivalent in composition at the start, why is the initial CFU/ml value 50-100 fold lower in B than in A? The text (line 93) refers to the “large fraction” of cells that survived. Whereas the fact that any of the cells survived is of great interest, a fraction of 1/1,000,000 can't be described as “large”. Line 102: Modify to “some non-sporulating cells still survived...”. For part C, it would be very informative to know the change in viability for the WT strain under these conditions. Showing that 80% of the living cells were spores at time X doesn't indicate what fraction of the original population sporulated. In other words, did the percentage of spores in the population increase because a large number sporulated or because the vast majority of the cells died?

In Fig. 3, are stationary phase cells the immediate precursors of the time 0 cells? If so, why are exponential phase cells provided as the control? It would be clearer to compare the starved cells to the stationary phase cells before they were filtered.

It seems surprising that the majority of genes up-regulated by starvation are carbon uptake and metabolism genes, with no indication of proteolytic enzyme induction or peptide or amino acid uptake.

Line 177: Do the results show that 135 genes were up-regulated both in “normal” stationary phase cells and in the 14-day starved cells compared to exponential phase cells? Similarly, do the data indicate that 115 genes were up-regulated in exponential phase and 14-day cells compared to stationary phase cells? Whether correct or not, please clarify.

Minor items:

1) The composition of various media is unclear. For Spizizen minimal medium, no reference is cited. This is important because the medium has a number of variabilities. In addition, the item listed as “Na₃C₆H₅O₇” is likely trisodium citrate, but for most readers it would be helpful to say so. Also, iron citrate is usually a mixture of FeCl₃ and Na₃-citrate, not Fe-NH₄-citrate. LB has many forms as well, so a clearer definition is needed. For example, the original medium contained glucose, but glucose is not usually used for *B. subtilis* cultures in LB.

2) Line 320: Change “none” to “non”.

3) Line 331: Change “wide spread” to “widespread”.

4) Line 389: Change “noise” to “noisy”.

5) Table 1: Change the lead sentence as follows: “The transcriptome profile of 14-day starved cells was compared to those of exponential phase and stationary phase cells.”

Reviewer #3 (Remarks to the Author):

Gray et al. report about interesting observations that suggest a novel state of dormancy in starved cells of *Bacillus subtilis*. They coin this extremely slow growth oligotrophic growth and suggest that

it may be relevant to other bacteria as well.

Major concerns:

#1 The use of a non-sporulating *spoIIE* mutant strain brings the questions whether the results are relevant for wild type bacteria. Which fraction of wild type bacteria would enter oligotrophic growth?

#2 l. 93: In contrast to the authors' statement, only a very small fraction of the cells remains viable, the cell count drops to 0.001%!

#3 l. 110/111, Fig. 1C: Please make a statement about the viability of the cells in the non-sporulating fraction. From the presented results, it is not clear whether the data obtained with the *spoIIE* mutant can be generalized!

#4 l. 417/418: The conclusion is not valid as general as made by the authors. They studied only a sporulation mutant, not the wild type.

Specific comments:

#1 l. 68: better "continues"

#2 l. 102/103: "Non-sporulating cells still survived", see above major #2: Taking into account the low frequency of survivors, this conclusion is not justified. The description gives a wrong impression here!

#3 l. 146/147, Fig. 3E: Why is the membrane potential so low at day 1, and by which mechanism could it increase until day 7?

#4 l. 179: What is the *areE* gene? Do you mean *araE* or *aroE*?

#5 SigL is not a cold-shock sigma factor!

#6 l. 307 ff: The discussion is quite long (nearly five pages). Please shorten to three pages.

Point-by-point reply to reviewers comments

Firstly, we would like to thank the reviewers for taken the time to carefully review our manuscript and to provide us with constructive comments. For clarity, we have indicated our answers in blue.

Reviewer #1 (Remarks to the Author):

This ms. describes interesting observations of the growth of *B. subtilis* under extremely nutrient poor conditions. Given that "natural" environments are more like these conditions, this work is potentially of broad interest and importance. However, many of the claims made by the authors are not sufficiently supported by the experiments described which greatly reduces my overall enthusiasm. In addition, there are a number of confusing statements about the data and important statistical tests and/or experimental details are missing.

1. How can the authors state "after 100 days a large fraction of cells were viable" (line 93), but the data in Fig. 1A show that less than 1 in 10x5 cells made cfus?

We agree with the reviewer. This wording stems from our experience with bacterial survivability (e.g. after antibiotic treatment) that is commonly depicted in log scales. We have now rephrased this sentence (line 93: "*but even after 100 days the culture still contained viable cells that formed colonies*").

2. What is the statistical significance of the differences in Fig. 1B? T60 does not look significant. How many biological replicates were conducted?

The key point of this experiment was to show that pure water, instead of buffer, still gave viable cells. The graph in Fig. 1B is based on three biological replicates. An unpaired two-tailed t-test, indicated that the difference between the buffer and water samples becomes statistically significant ($p < 0.05$) after time point 'day 7'. We have added this information now to the legend of Fig. 1B (line 738).

3. Under these growth conditions, did they confirm that the pH of medium did not change? Certainly with LB, it needs to be buffered for growth >24h

For LB that is indeed true, but we have not used LB and we grew all our cultures in minimal SMM medium that contains a phosphate buffer. The 'starvation buffer' contained the same phosphate buffer (100 mM in both cases). The composition of SMM medium and the starvation buffer is described in the Method section.

To be absolutely sure that the pH remained stable, we followed the reviewer's suggestion and repeated the experiment in starvation buffer (2 independent experiments) and measured the pH after 0, 4, 7, 11, and 14 days. Indeed, we could not detect any change in pH, confirming that the phosphate buffer was sufficient to keep pH stable. We mention this now in the text in line 104.

4. It is not true that sporulation "cannot be reversed once started" (line 62) –sporulating cells are not committed to growth until after asymmetric division (Parker & Errington, 1996) when a sigF gene is expressed (Dworkin & Losick 2005). Furthermore, while spoIIE mutants are unable to complete sporulation, they can initiate sporulation, and in fact the phenotype of a spoIIE null mutant is that it is disporic (Illington & Errington 1991).

We thank the reviewer for pointing this out, and we have now changed the sentence to "cannot be reversed once the asymmetric sporulation septum has been formed" (line 62), and we have added the Parker and Dworkin references.

However, whether the *spoIIE* mutant forms disporic cells is not relevant here. The key point is that the mutant does not make viable spores. Actually, according to Illington & Errington 1991, the fraction of disporic cells is the same as in wild type cells (Fig. 1C, morphology class 3, in their paper). Aside of this, we have tested several other sporulation

mutants in the later genetic experiments and they all gave the same results as the *spoIIE* mutant ($\Delta spo0A$, $\Delta sigE$, -F, -G, -H, -K in Figs 8D and S8).

5. In their analysis of wt cells (Fig. 1C), they interpret their data as showing that a substantial fraction of the population are non-sporulating cells. This interpretation depends on the assumption that the spore cannot germinate under these conditions. Do we know this? A germination defective mutant might be useful, in this respect. There is quite possibly a dynamic equilibrium between spores and non-spores, which would not be revealed by discontinuous sampling (as opposed to continuous observation).

The reviewer raises here an interesting point; whether during long periods of starvation there is a cycle of germination and sporulation. This is in theory possible since *B. subtilis* spores can germinate spontaneously even when conditions are unfavourable, albeit at low frequencies (Sturm & Dworkin 2015). To address this question, we followed the viable count of a germination mutant (strain FB113) that makes spores which cannot germinate (Paidhunat & Setlow 2000). During the first 11 days of starvation, this strain followed a reduction in viable count comparable to the *spoIIE* mutant. After that the viable count decreased further and leveled out at approximately 10^4 CFU/ml, but never approached zero. Thus, cells can still form spores in the first few days of starvation, but there is no continuous germination-sporulation cycle going on. We have now added this experiment to the manuscript (lines 117, Fig. S3).

6. In order to make claims about the metabolic state of these cells, then they must directly assay parameters such as DNA replication, RNA synthesis, or protein synthesis using incorporation of radiolabeled precursor. This will give at least an average level of these processes. To definitively show that most of the cells are active (as opposed to only a subset), they must use fluorescent probes, so that they can image single cells.

We disagree with the reviewer on this point. We show in 3 completely different ways that starved cells are metabolically active. Firstly, we show that they are sensitive to the membrane potential dissipating antibiotic valinomycin, and we show that the vast majority of cells have a clear membrane potential. Since the cell membrane is not 100% ion impermeable (e.g. weak acids will leak through), and since it is highly unlikely that cells have shut down all transporters, including the ATPase synthase that consume the membrane potential, the presence of a membrane potential is only possible when cells are metabolically active and undergo respiration. Secondly, the induction of GFP synthesis indisputably shows that starved cells can still synthesise proteins, which is a highly energy-consuming process only carried out by metabolically active cells. Thirdly, starved cells grow longer in the presence of 3-MBA, which, again, is only possible when they are metabolically active and able to produce the energy, precursors and proteins required for cell growth. Because of these reasons, we have not performed radiolabelled precursor incorporation experiments.

7. The GFP experiment (Fig. 4) has too many caveats to be interpreted as a strong indicator of protein synthesis. For example, how do they know that the inability to synthesize GFP is not due to an inability to synthesize the *gfp* mRNA?

The key result in Fig. 4 is that cells that have been starved for a long time (7 to 14 days) are able to synthesise GFP and thus undergo both transcription and translation. Only for time point 'day 0' do we not see a strong induction of GFP, which might or might not be due to lack of transcription capability. In fact, 'day 0' cells also show a lower membrane potential (Fig. 3E). The experiments shown in Fig. 8A suggest that this is due to the absence of nutrients in the medium after the stationary phase cells were filtered and resuspended into fresh medium. We have now added this explanation to the main text (line 255).

8. Their indirect assay for growth (Fig. 7) depends on the ability of the 3-MBA compound to block FtsZ polymerization. Since this assay depends on an inhibitor, they need to perform experiments with the FtsZ mutant that is no longer 3-MBA sensitive to ensure that their action of the compound under their experimental conditions is specific to FtsZ. They

also need to explain why they are not using other methods which do not depend on inhibitor (such as the TIMER system of Claudi et al 2014).

We agree that this is indeed an important control and we have now repeated the 3-MBA experiment with an *ftsZ* mutant strain (FtsZ-G196S,N263K) that is resistant to this compound (Adams & Errington 2016). Indeed, this mutant was indeed unable to form long cells in the presence of 3-MBA under long term starvation conditions, in line with our expectations. We describe this control now in line 236 and Fig. S6.

The TIMER system described by Claudi et al. 2014 is based on a fluorescent DsRed variant that spontaneously changes colour from green to green/orange. This system uses the dilution rate of these fluorescent variants during cell growth as a measure of cell age, and can measure differences in generation times in the range of several hours. However, we are dealing with doubling times of days and it is questionable whether the dilution during oligotrophic growth is sufficient to use the TIMER system. Moreover, the maturation and stability of the DsRed variants might be different in our system, which could affect the interpretation of the results. Finally, the TIMER system, although elegant and smart, is an indirect method to follow growth. Our 3-MBA method is a direct and unambiguous detection of cell growth.

9. I do not understand their discussion of noise (line 389-395) – how are they measuring noise in their system – is it intrinsic or extrinsic – the data in Fig.4 are not sufficient to make any specific claims.

We agree that our GFP expression measurements of Fig. 4 are not indicative of the expression noise that we discuss in this paragraph. Therefore, we have removed the reference to this figure (line 414).

Reviewer #2 (Remarks to the Author):

The manuscript presents a very broad and very detailed analysis of the impact of carbon starvation on *B. subtilis* cells (adequate nitrogen, phosphorus and sulfur continued to be provided), with most of the data obtained with non-sporulating mutant strains. This approach makes the results of broader relevance to bacterial species, but does limit the interpretation for *B. subtilis* itself. Nonetheless, the results are intriguing and will undoubtedly surprise the majority of readers.

In several cases, the information provided is insufficient to provide a full understanding of what has happened. For instance, in the paragraph beginning on line 94, the reader is not informed as to whether the optical density of long-term cultures was measured as a potential indicator of cell lysis. Also, the statement that starvation resulted in “release of more nutrients” is not supported by any data provided.

Indeed, in the first days we observed by eye a strong drop in OD. We agree that it is useful to mention this in the main text and have added a sentence describing this (line 97). However, OD measurements cannot reliably detect culture densities at low cell numbers. Therefore, we have used viable counts instead of OD measurements in our starvation experiments.

Fig. 1: The legend refers to CFU but doesn't name the medium used.

We have now added the reference to medium (SMM medium) above Fig. 1A.

In addition, it would be very helpful to include in parts A and B the OD values at each time point.

Due to relative low cell numbers, OD measurements cannot be used to follow cell viability.

If the samples in parts A and B were equivalent in composition at the start, why is the initial CFU/ml value 50-100 fold lower in B than in A?

In the experiment in Fig. 1B, we filtered cells after an initial 2 days of growth that

comprises an initial log phase ($1/3$ day) followed by a substantial lag phase ($1^{2/3}$ days). Fig. 1A shows that during these 2 days, the CFU goes down by almost 2 log units. That is why the starting CFU number in B corresponds to the CFU number at time point '2 days' in graph A. We have explained this now in the legend of Fig. 1 (line 736).

The text (line 93) refers to the "large fraction" of cells that survived. Whereas the fact that any of the cells survived is of great interest, a fraction of $1/1,000,000$ can't be described as "large". Line 102: Modify to "some non-sporulating cells still survived...".

We agree with the reviewer. This wording stems from our experience with bacterial survivability (e.g. after antibiotic treatment) that is commonly depicted in log scales. We have now rephrased this sentence (line 93: "*but even after 100 days the culture still contained viable cells that formed colonies*").

For part C, it would be very informative to know the change in viability for the WT strain under these conditions. Showing that 80% of the living cells were spores at time X doesn't indicate what fraction of the original population sporulated. In other words, did the percentage of spores in the population increase because a large number sporulated or because the vast majority of the cells died?

This is a good point. The absolute number of spores increases steadily over time, but indeed, the strong increase in the first days is largely due to the decrease in non-sporulating cells. To make this clear, we have plotted the absolute CFU counts of spores and cells and added an inset depicting the percentage of spores (new Fig. 1C). We have moved the biological replicates and the experiments with pure water to the supplementary information (new Fig. S2).

In Fig. 3, are stationary phase cells the immediate precursors of the time 0 cells? If so, why are exponential phase cells provided as the control? It would be clearer to compare the starved cells to the stationary phase cells before they were filtered.

The log phase cells were used as a positive control to show that the antibiotics are actually working. Comparing cells immediate before and after filtering was not the purpose of this experiment. The take home message is that cells that have been starved for several days to weeks are tolerant to most antibiotics but still sensitive to valinomycin.

It seems surprising that the majority of genes up-regulated by starvation are carbon uptake and metabolism genes, with no indication of proteolytic enzyme induction or peptide or amino acid uptake.

There is a significant but very moderate (1.2- to 1.5-fold) upregulation of the protease genes *aprE* and *nprE*, the peptide permease operon *dppBCDE* and the amino acid transporter genes *rocC* and *hutM* when 14 day culture samples are compared to log phase cultures. However, this level of regulation was insufficient (less than 2x) to discuss in the paper. Possibly, the limited induction is related to the fact that cells were grown in minimal medium and therefore able to synthesise their own amino acids. Alternatively, this minimal medium might have triggered the induction of transporters even in log phase. It might also have to do with the fact that high induction of proteases and transporters is not necessary when cells are growing very slowly. As we show in Fig. 8F, the presence of a single protease is sufficient to survive oligotrophic conditions.

Line 177: Do the results show that 135 genes were up-regulated both in "normal" stationary phase cells and in the 14-day starved cells compared to exponential phase cells? Similarly, do the data indicate that 115 genes were up-regulated in exponential phase and 14-day cells compared to stationary phase cells? Whether correct or not, please clarify.

We are a bit confused by the numbers. In line 177 we wrote: "The number of genes that were at least 2-fold upregulated in the 14 days starved culture compared to the exponentially and stationary phase cultures, were 165 and 145 genes, respectively (Fig. 5C).".

We have tried to make this more clear by rephrasing this as follows: "165 genes were at least 2-fold upregulated in the 14 days starved culture when compared to the

exponentially culture, and 145 genes were at least 2-fold upregulated in the 14 days starved cultures when compared to the stationary phase cultures." (line 191).

Minor items:

1) The composition of various media is unclear. For Spizizen minimal medium, no reference is cited. This is important because the medium has a number of variabilities. In addition, the item listed as "Na₃C₆H₅O₇" is likely trisodium citrate, but for most readers it would be helpful to say so. Also, iron citrate is usually a mixture of FeCl₃ and Na₃-citrate, not Fe-NH₄-citrate. LB has many forms as well, so a clearer definition is needed. For example, the original medium contained glucose, but glucose is not usually used for *B. subtilis* cultures in LB.

This is a good point. We have added the original reference for SMM medium and we have better described the chemical composition of SMM and LB in the Methods (line 459).

2) Line 320: Change "none" to "non".

Done.

3) Line 331: Change "wide spread" to "widespread".

Done.

4) Line 389: Change "noise" to "noisy".

Done.

5) Table 1: Change the lead sentence as follows: "The transcriptome profile of 14-day starved cells was compared to those of exponential phase and stationary phase cells."

Corrected as suggested.

Reviewer #3 (Remarks to the Author):

Gray et al. report about interesting observations that suggest a novel state of dormancy in starved cells of *Bacillus subtilis*. They coin this extremely slow growth oligotrophic growth and suggest that it may be relevant to other bacteria as well.

Major concerns:

#1 The use of a non-sporulating spoIIE mutant strain brings the questions whether the results are relevant for wild type bacteria. Which fraction of wild type bacteria would enter oligotrophic growth?

In Fig. 1C we followed the viability of cells and spores in a wild type culture. To make it more clear which fraction survives, we have now including the CFU of spores and cells (new Fig. 1C and Fig. S2). Approximately a quarter of the population appears to enter oligotrophic growth (line 114). See also reply to remark #3 below.

#2 l. 93: In contrast to the authors' statement, only a very small fraction of the cells remains viable, the cell count drops to 0.001%!

We agree with the reviewer. This wording stems from our experience with bacterial survivability (e.g. after antibiotic treatment) that is commonly depicted in log scales. We have now rephrased this sentence (line 93): "*but even after 100 days the culture still contained viable cells that formed colonies*".

#3 l. 110/111, Fig. 1C: Please make a statement about the viability of the cells in the non-sporulating fraction. From the presented results, it is not clear whether the data obtained with the spoIIE mutant can be generalized!

We are a bit confused with what the reviewer means with "viability of cells in the non-sporulating fraction". By measuring CFUs, we automatically use 'viability' of cells. In

other words, the non-sporulating fraction is the viable non-sporulating fraction. Possibly this confusion arises from the fact that we depicted only the percentage of spores. To clarify this, we have now added CFUs numbers of spores and cells (non-spores) in Fig. 1C (see also Fig. S2).

#4 l. 417/418: The conclusion is not valid as general as made by the authors. They studied only a sporulation mutant, not the wild type.

This is a good point and we have now performed the 3-MBA growth experiment also with wild type cells. This gave similar results as those obtained with the *spoIIE* mutant. We have added these findings to the manuscript (line 235, new Fig. S6). Therefore, it is reasonable to assume that our conclusions are also applicable to wild type Bacilli.

Specific comments:

#1 l. 68: better "continues"

Corrected.

#2 l. 102/103: "Non-sporulating cells still survived", see above major #2: Taking into account the low frequency of survivors, this conclusion is not justified. The description gives a wrong impression here!

We have adjusted the text as described in the reply to major #2 .

#3 l. 146/147, Fig. 3E: Why is the membrane potential so low at day 1, and by which mechanism could it increase until day 7?

Presumably, this is because the cells have been filtered and resuspended into fresh buffer without any nutrients, and not sufficient cells have yet lysed to provide these nutrients. We have added this explanation to the text in line 255.

#4 l. 179: What is the *areE* gene? Do you mean *araE* or *aroE*?

This should indeed have been *araE*. We have corrected this.

#5 SigL is not a cold-shock sigma factor!

This is indeed incorrect. The SigL regulon is not induced by cold shock, but a *sigL* mutant is cold sensitive (Wiegeshoff, Beckering, Debarbouille, Marahiel, 2006). We have corrected this as follows "or the sigma factor SigL, which renders cells cold-sensitive" (line 277).

#6 l. 307 ff: The discussion is quite long (nearly five pages). Please shorten to three pages.

Our findings give plenty to discuss. Since the other reviewers had no objections to the length of the discussion, we would like to keep it as it is.

Reviewers' comments:

Reviewer #1 (Remarks to the Author):

The authors have done an appropriate job in answering most of my critique. However, their answer to #6 is not acceptable. All of their measurements (3 different ways) are indirect; I feel strongly that a direct measurement of metabolic activity is necessary. There are numerous ways to do this, either with radioactive or fluorescent derivatives of DNA/RNA precursor or amino acids, so there is really no reason that the authors should not be expected to do at least one direct measurement.

Reviewer #2 (Remarks to the Author):

The revised version of the manuscript includes critical experiments that answer some of the major questions previously raised by reviewers. The discoveries are of great interest and value, but the text still needs some modifications in order to be clear and correct.

1) Line 114 and Fig. 1: Describing the non-sporulating population as 20-30% based on CFU with and without heating can be misleading. That is, spores can form colonies without having been heated, although their efficiency of colony-formation is lower. Therefore, the CFU in the unheated population is not necessarily only from non-sporulating cells. A less ambiguous analysis would be to simply examine the population in the microscope. Doing so would reveal spores, vegetative cells and cells in the process of spore formation.

2) Line 172: The authors cannot rule out the possibility that the 14-day old cells take up IPTG better than day 0 cells.

3) Line 181: The levels of protein have not been measured. It is the proportion of cells that make GFP that is being measured, not the amount of GFP per cell. Since the viability decreased about 50-fold, the cells producing GFP may have been the ones that survived better.

Smaller issues:

1) Lines 38,39: It would be better to refer to "Some genera, such as Bacillus, Streptomyces and Clostridium,..."

2) Line 45: It would be helpful to readers to know that the cited papers revealed 10,000-fold decreases in populations of Flavobacterium and Brucella.

3) Line 93: The readers would be helped by being told that the remaining viable cells represented 1/100,000 of the original population.

4) Fig. S2: The legend refers to part C; it is actually part B.

5) Line 160: Change "lead" to "led"

6) Line 288: Change "comparably" to "comparable"

Reviewer #3 (Remarks to the Author):

The authors did a good job with their revision.

Point-by-point reply to reviewers comments

Firstly, we would like to thank the reviewers for taken again the time to review our revised manuscript. For clarity, we have indicated our replies to their comments in blue.

Reviewer #1 (Remarks to the Author):

The authors have done an appropriate job in answering most of my critique. However, their answer to #6 is not acceptable. All of their measurements (3 different ways) are indirect; I feel strongly that a direct measurement of metabolic activity is necessary. There are numerous ways to do this, either with radioactive or fluorescent derivatives of DNA/RNA precursor or amino acids, so there is really no reason that the authors should not be expected to do at least one direct measurement.

We have now added three new experiments to provide extra evidence that oligotrophic cells are metabolically active. Firstly, we performed nascent protein incorporation assay using fluorescent amino acid analogue as suggested by the referee (Fig. S6A&B, line 178 main text). Secondly, we used fluorescent precursor incorporation to show peptidoglycan synthesis (Fig. S6C, line 186 main text). Thirdly, we used DNA staining to reveal ongoing DNA synthesis in 3-MBA treated cells (Fig S7B, line 251, main text). These experiments further supported our conclusion that these cells are metabolically active.

Reviewer #2 (Remarks to the Author):

The revised version of the manuscript includes critical experiments that answer some of the major questions previously raised by reviewers. The discoveries are of great interest and value, but the text still needs some modifications in order to be clear and correct.

1) Line 114 and Fig. 1: Describing the non-sporulating population as 20-30% based on CFU with and without heating can be misleading. That is, spores can form colonies without having been heated, although their efficiency of colony-formation is lower. Therefore, the CFU in the unheated population is not necessarily only from non-sporulating cells. A less ambiguous analysis would be to simply examine the population in the microscope. Doing so would reveal spores, vegetative cells and cells in the process of spore formation.

We have followed the reviewer's advice and repeated this experiment up to 14 days starvation and assessed sporulation by light microscopy. This gave comparable ratios for spores and non-sporulating cells (new Fig. S2B, line 116 main text).

2) Line 172: The authors cannot rule out the possibility that the 14-day old cells take up IPTG better than day 0 cells.

In the literature, we have not found any indication that IPTG uptake varies between log phase and stationary phase *Bacillus* cells. Even if this is the case, it is not

relevant for the main conclusion of this experiment, as the aim of this experiment was to show that 14 days starved cells are still able to induce protein expression.

3) Line 181: The levels of protein have not been measured. It is the proportion of cells that make GFP that is being measured, not the amount of GFP per cell. Since the viability decreased about 50-fold, the cells producing GFP may have been the ones that survived better.

We do not understand this remark. We have measured GFP levels in individual cells using fluorescence light microscopy. The cells were subsequently 'binned' in intervals of increasing fluorescence, as indicated on the Y-axis in Fig. 4. So we have actually measured protein levels for individual cells. To be clearer, we have explained the binning procedure in the legend of Fig. 4 (line 816 main text).

Smaller issues:

1) Lines 38,39: It would be better to refer to "Some genera, such as Bacillus, Streptomyces and Clostridium,..."

This is indeed a more accurate wording and we have adjusted the text accordingly.

2) Line 45: It would be helpful to readers to know that the cited papers revealed 10,000-fold decreases in populations of Flavobacterium and Brucella.

Since the measure of decrease depends on time and conditions, we prefer to use a more general wording, and we have now added the following sentence: "It should be mentioned that in all these cases it was only a small fraction of the population that survived." (line 47, main text).

3) Line 93: The readers would be helped by being told that the remaining viable cells represented 1/100,000 of the original population.

To us, this number is clear from the graph in the figure (Fig. 1A). However, we have now rephrased the conclusion by mentioning that: "even after 100 days the culture still contained some viable cells" (line 94 main text).

4) Fig. S2: The legend refers to part C; it is actually part B.

This has now been corrected.

5) Line 160: Change "lead" to "led"

This has now been corrected.

6) Line 288: Change "comparably" to "comparable"

This has now been corrected.

Reviewer #3 (Remarks to the Author):

The authors did a good job with their revision.